# Nutrition Supports Deconstructed and Disrupted: An Evaluation of a Multilevel School-Based Intervention during the Time of COVID

**DOI:** 10.3390/ijerph182111006

**Published:** 2021-10-20

**Authors:** Rachael D. Dombrowski, Bree Bode, Kathryn A. G. Knoff, James Mallare, E. Whitney G. Moore, Noel Kulik

**Affiliations:** Division of Kinesiology, Health and Sports Studies, College of Education, Wayne State University, Detroit, MI 48202, USA; bodebr@wayne.edu (B.B.); kathryn.gray@wayne.edu (K.A.G.K.); jpmallare@wayne.edu (J.M.); whitneymoore@wayne.edu (E.W.G.M.); ab7564@wayne.edu (N.K.)

**Keywords:** food security, COVID-19, low income, health inequities, evaluation, intervention, schools

## Abstract

The Best Food Forward (BFF) project aims to provide multiple nutrition supports and interventions to improve family food security (FS) and health outcomes associated with FS within two metropolitan school districts. A quasi-experimental time-series design guided a multilevel evaluation for BFF through surveys, biometric screenings, focus groups, and observations among a random sample of caregiver–child dyads. FS, utilization of school meal programs, and nutrition behaviors were observed and analyzed at three time points: preintervention, postintervention pre-COVID-19, and postintervention post-COVID-19. Participants included 122 parents and 162 youth. Families reported (1) an income less than $35,000 annually (48.8%) and (2) a COVID-19-related job loss (36.9%). Parents used Supplemental Nutrition Assistance Programs or Women, Infants, Children benefits prior to (51.1%) and following COVID-19 (50.0%). No significant differences in FS were found. RM-ANOVA indicated an increase in breakfast consumption at home and a decrease in use of the school breakfast program (*F*(1.78, 74) = 19.64, *p* < 0.001, partial *η*^2^ = 0.21) and school lunch program (*F*(1.51, 74) = 23.30, *p* < 0.001, partial *η*^2^ = 0.24). Rates of FS and eating behaviors did not change significantly over time. Correlations of program usage and eating behaviors demonstrate the importance of promoting participation in school meal programs. BFF may have prevented significant decreases in FS during COVID-19.

## 1. Introduction

A significant proportion of populations throughout the globe experienced food insecurity prior to the COVID-19 pandemic (687.7 million) and the rate of food insecurity is projected to increase (>800 million people) through 2030, regardless of the effects of the pandemic [1,2,3]. In the United States (U.S.), food insecurity is defined as a lack of consistent access to enough food for an active, healthy life [4]. Prior to March 2020, 35 million people reported some level of food insecurity in the U.S. [5], and projections indicate that the number of food-insecure people will increase to 42 million by the end of 2021 (1 in 8 individuals and 1 in 6 children) due to the effects of the COVID-19 pandemic [5].

Access to, availability of, and consumption of nutritious foods are key components of food security [1]. Policies, systems, and environments (PSEs) within communities, states, and countries can either support or hinder access to, availability of, and consumption of healthy foods; therefore, they have a direct effect on the food security of local populations [4]. This was especially evident during the COVID-19 pandemic when additional nutrition and economic supports were provided to low-income communities throughout the U.S. and steadied food insecurity rates despite increased job and income loss among millions of families [1,2,3,4,5]. PSEs that are not formulated with a health equity lens have often perpetuated food insecurity among low-income and racial and ethnic minority communities, especially in the United States [6]. This has caused disproportionate rates of (1) unemployment, (2) poverty, (3) impacts on health across generations, and (4) poor health behaviors among primarily African American/Black and Latinx/Hispanic populations and has continued to contribute to health inequities among communities of color [7,8,9,10].

### 1.1. Food Security, Unemployment, and Poverty Rates

Due to COVID-19, families in the U.S. experienced unemployment (14.8%) and underemployment at the highest rates recorded since the Great Depression [7]. This was most often experienced by Hispanic/Latinx and African American/Black communities who were employed in many of the hospitality and travel industries that were closed during the height of the pandemic (Table 1) [11,12,13]. In May 2020, women also (13.9%) reported a higher unemployment rate than men (11.9%) [13,14,15,16,17,18]. The loss of income in many households increased poverty rates and, in turn, food insecurity among families and children as communities responded to the COVID-19 pandemic, and subsequent closures in schools resulted in a lack of participation in school meal programs and other nutrition supports (Table 1) [13,18,19,20].

### 1.2. Food Security, Systemic and Institutionalized Racism, and Health Behavior

Systemic and institutionalized racism within the U.S. has burdened communities with poor health outcomes across familial generations of African American/Black and Latinx/Hispanic populations [21,22,23,24]. Due to segregation, discrimination, and inequitable policies, low-income racial and ethnic minority communities have limited opportunities for intellectual, financial, or physical wellness; furthermore, they have experienced little to no access to quality education, nutrition, and viable healthcare [22,25,26,27]. Education is a crucial determinant of health. Individuals withheld from formal schooling due to sociopolitical restraints such as historical segregation in school systems, economic hardship, and geography are more likely to have low employment rates and greater incidence of food insecurity later in life [28,29]. In addition, food insecurity has become a pressing public health challenge consistent with child development [30]. Familial exposure to community violence, negative social relationships, economic hardship, and Adverse Childhood Experiences (ACEs) impact accessibility and choice of food sources [31]. The relationship between sustained ACEs (sexual abuse, home-insecurity, alcoholism, etc.) and food insecurity creates a concerning pattern of poor health outcomes among children, especially when the household is led by a mother. In these cases, child hunger multiplies, and food insecurity is recognized as a root problem in emotional and cognitive youth development [21,32,33,34,35,36].

Positive nutrition-related behaviors and physical activity can moderate the negative effects of stress, including the negative stress accumulated from poverty and/or food insecurity [37,38]. Negative stress is a significant burden among school-aged youth navigating food insecurity, which can impede learning and social–emotional development and is implicated in chronic disease progression [38,39,40]. People who have experienced food insecurity may make poor nutrition-related choices out of the learned, complex reality of scarcity [41]. The physiological response, under the constraints of scarcity, is to fulfill the need to consume calories, even empty calories (e.g., soda, fast food) when food becomes available [41]. Youth that live in food-insecure households have health habits and outcomes that are linked to increased rates of overweight and obesity due to this physiological response [21,42,43]. In addition, the COVID-19 pandemic introduced new stressors to already food-insecure families and more U.S. families experienced the stress of food insecurity due to the pandemic and accompanying loss of income [44,45].

### 1.3. Food Security and Health in the Time of COVID-19

Rapidly spreading communicable diseases (in this case, COVID-19) are relationally reciprocal to food-insecurity rates [46,47,48]. As food accessibility is decreased, immunity from proper nutrition also decreases, resulting in increased susceptibility to morbidity and mortality from communicable disease; a mounting health inequity [46]. In addition, higher communicable disease infection and death rates increase food insecurity especially for frontline/essential workers who are typically uninsured, underinsured, or have no paid time off benefit causing a loss of wages due to unpaid time off [46]. Households in which the primary wage earner contracted COVID-19 and/or died contributed to food insecurity of family members left behind, which resulted in food insecurity experienced by family members outside the peaks of the pandemic [46].

Lasting effects of the reciprocal relationship between COVID-19 and health outcomes that might also extend beyond the peaks of the pandemic are those associated with mental health [46]. Wolfson, Garcia, & Leung (2021) found that one-third of the adults they screened for food insecurity during COVID-19 also screened positive for depression (33%), anxiety (39%), and high stress (39%). Consequences of food insecurity are also linked to chronic diseases and conditions—namely, hypertension, coronary heart disease (CHD), hepatitis, stroke, cancer, asthma, diabetes, arthritis, chronic obstructive pulmonary disease (COPD), and kidney disease [46]. The SARS-2 virus-infected individuals who often had a health diagnosis or previously had a depreciating health event including but not limited to the 10 typical noncommunicable diseases associated with food insecurity [44,45,46].

### 1.4. The Initiation of Federal, State, and Local Solutions Pre and Post the Onset of COVID

The impacts of COVID-19 were evident in rapid response actions taken by federal, state, and local officials to change social policies, unemployment rates, and supply chain demand and interruptions [5,14,49]. Families experienced a loss of social, cognitive, and nutrition supports as typical environments (schools, places of worship, community centers, etc.) temporarily, and in some cases indefinitely, closed [14,49]. Measures were taken to moderate social policies that would mediate federally funded nutrition programs and the impact of the COVID-19 pandemic on access to, availability of, and utilization of food programs among both adults and children in the U.S. This included the expansion of federal nutrition support programs, flexibility within the school lunch and breakfast programs, and the distribution of the Pandemic Electronic Benefits Transfer Program (P-EBT) to parents of school-aged children [5]. The rapid response policy changes made it possible for some of the disproportionately affected communities in the U.S. to have more access and availability to nutrition supports and eased some of the burdens of food insecurity [5,24,50]. For example, the federal funding provided made it possible to offer school meal programs to youth at their homes via bus routes or local pick-up and ensured that all children were being fed at home regardless of their school attendance status [5,41]. However, many of these circumstantially created benefits will expire in 2021 with no tentative plan for continuation. The effects of the expiration of these benefits on food-insecure households must continue to be assessed and reported.

### 1.5. Food Security in Michigan and the Best Food Forward Intervention

Michigan was not immune to the effects of the COVID-19 pandemic. Nearly 1.5 million Michigan residents experienced some form of food insecurity (May 2020) during the pandemic as compared with 1.2 million Michigan residents pre-pandemic (February 2020) [49,50]. To further describe the impact of food insecurity pre-COVID, in February 2020, Michigan’s 1.2 million residents received $137 million in food assistance, while in May of 2020, residents received $263 million in food assistance [51]. Wayne County, which comprises Detroit, was one of the hardest hit areas in the state [5]. The unemployment rate in Michigan during the height of the pandemic was 24%, much higher than nationally reported rates (15%) [52,53,54,55]. Similar to national trends, African American/Black Michiganders experienced unemployment at double the rates of Whites [52,56]. Additionally, Black/African American and Latinx/Hispanic families experienced poverty at three and two times the rate of Whites, respectively, in Michigan [56].

Prior to the onset of COVID-19, the Michigan Department of Education (MDE) worked closely with school districts to introduce and encourage the utilization of Universal Free Lunch, when applicable, to remove and reduce barriers to utilization of the school meal programs [57]. During the pandemic, MDE worked collaboratively with school districts to help with the interpretation of federal policies and adjustments to school meal programs that were provided flexibility to ensure school meals could be distributed to all school-aged children in the community, regardless of the family’s economic status [57]. In 2019, the MDE brought local food banks, community partners, and school districts in two communities together to develop and implement the Best Food Forward (BFF) pilot program as an opportunity to bring families to full food security.

Best Food Forward is an 8–10-year initiative to bring children of two school districts to full food security in order to achieve better outcomes in academic achievement, attendance, behavior (fewer discipline referrals), mental and physical health, and household stability. The goal is to provide food security for each child and their household at school, at home, and while in their community by the following methods:

Ensuring the district is fully utilizing all federal child nutrition programs;Launching regular mobile food pantries at schools;Engaging family members, administrators, teachers, and the community;Working to increase and coordinate existing nutrition education programs in schools;Providing information on and increasing the use of available community services; andConducting community engagement in order to identify new models of food delivery to meet the needs of those that current tools do not satisfy.

The BFF model is a pilot to understand what can be scalable and replicable for other districts across the State of Michigan. The purpose of the evaluation was to determine the impact of the BFF intervention through a multilevel assessment. It was hypothesized that the Best Food Forward intervention would relieve food insecurity first among students and second among their households by (1) increasing food access and food availability through mobile pantries, (2) increasing awareness and utilization of the school meal programs, and (3) developing relationships between community food banks and school district staff to coordinate nutrition education that would benefit the local families. The original hypothesis did not account for the impact of the COVID-19 pandemic.

This paper will introduce the BFF intervention and report on the baseline assessment impact of the intervention within the context of the COVID-19 pandemic. This paper will include the impact of the intervention on food security of participating families, their participation in school meal programs, and nutrition outcomes. In addition, recommendations for policy and systemic changes that can be implemented with federal, state, and local partners will be discussed.

## 2. Materials and Methods

### 2.1. Data Collection/Sampling

A quasi-experimental time-series design was used to obtain data from two school districts in two midwestern communities. The original evaluation plan included assessments of food security, nutrition, health, and academic outcomes within two socioeconomically similar school districts (SD1 and SD2) at four time points throughout the two-year pilot project. A randomly selected sample of 100 youth in 3rd, 6th, and 9th grades and their caregivers (hereafter referred to as parents) (*n* = 100) from each district was recruited for participation in the evaluation. Due to delays caused by COVID-19, baseline data among the parents were collected in both school districts between August 2020–June 2021. In total, 122 parents (SD1 *n* = 46; SD2 *n* = 73; no district identified *n* = 3) and 162 youth (SD1 *n* = 105; SD2 *n* = 57) who were enrolled in 3rd, 6th, and 9th grades in the 2019–2020 school year were randomly selected for participation in the BFF evaluation. Self-reported survey data was collected from each school district participant at baseline and included three embedded timeframes for the parent participants to report on retrospectively. The three timepoints are as follows: Prior to August 2019 (T0: preintervention; *n* = 90), August 2019–March 2020 (T1: postintervention, pre-COVID; *n* = 83), and March 2020–present (T2: post-intervention, post-COVID; *n* = 98). Intervention outcomes were measured through a variety of methods including in-depth interviews, focus groups, surveys, observations, and document review within the two school districts. In this report, results of the parent surveys will be discussed.

### 2.2. Instruments

Parent demographic information included identified race/ethnicity, gender, unemployment status pre- and post-COVID-19, income level, and age. Parents also reported food security status for T1 (postintervention, pre-COVID) and T2 (post-COVID). The survey instrument was modified to account for the effects of COVID-19 in June 2020. As this was meant to be a baseline measure for the entire evaluation, the food security and federal nutrition support participation variables were only assessed pre- and post-COVID (T1 and T2). Utilization of school meal programs, community food programs, and nutrition behaviors were assessed at all three timepoints to determine the effect of BFF intervention activities (T0, T1, and T2).

To measure how frequently the parents typically consumed fruits and vegetables, the following five items from the Youth Risk Behavior Surveillance Survey (YRBSS) from the Centers for Disease Control and Prevention [58,59] were used: the frequency at which they consumed 100% fruit juice, whole fruit, salad, potatoes, and other vegetables. They reported their consumption for a typical 7-day week during the T0 and T1 time periods, and over the course of the past seven days for T2 data collection. Response options ranged from “0 times in the last 7 days” to “4 or more times daily” providing response options 1, 2, 3, 4, 5, or 6 times in the last seven days to 1, 2, or 3 times daily on a 10-point scale. As completed in previous research, to put all responses on the daily intake metric, responses representing less than once per day (i.e., 1–6 times per week) were divided by 7 to determine an estimate for daily intake in servings [60,61].

The parent survey also assessed the use of different nutrition support programs at the school, community, state, and federal level. Specifically, use of school nutrition programs were assessed through measuring the frequency of weekly participation in school breakfast, school lunch, and after-school feeding programs (0–5 days of participation). Following the onset of COVID-19, after-school programs were no longer assessed in the survey; instead, use of the school lunch and breakfast programs (i.e., food delivered to homes or picked up from the school) was assessed. Utilization of food pantries or soup kitchens sponsored by either the school or their community were also measured. Additionally, federal and state program usage was assessed, including the use of Double Up Food Bucks (DUFB); awareness of DUFB; use of Supplemental Nutrition Assistance Programs (SNAP); and use of Women, Infants, Children (WIC).

To measure food security, researchers adapted the U.S. Department of Agriculture’s (USDA) shortened Household Food Security Survey for adults [62,63]. One item was not used on the adult survey as it asked a duplicate question [62,63]. Following USDA coding methods, responses were first coded as “affirmative” (1) or not (0), and then summed for a total score; these total scores were used to create the following food security status categories: 0–1 was high or marginal food security; 2–3 was low food security; and 4–5 was very low food security [63].

### 2.3. Statistical Analyses

Data were cleaned in Microsoft Excel prior to importing into SPSS v.27 (IBM, Armonk, NY, USA)) for further analysis. Frequencies and descriptive statistics were calculated for participants’ demographics, food security, eating behaviors, and use of school and community nutrition supports. The data were examined for outliers using boxplots and variables were tested for normality by examining skewness and kurtosis [64].

Pearson correlations determined if food security was related to the use of food programs. To determine the impact of the intervention on participants’ food security, a Wilcoxon Signed-Rank test comparing the categories of food security across the sample at T1 and T2 was conducted. To examine the effect of the intervention on parents’ use of food pantries and school programs, repeated-measures ANOVAs (rmANOVA) were conducted with data from all three timepoints. Finally, a rmANOVA was conducted to determine if participants’ daily fruit and vegetable intake had changed over time. Mauchly’s test of sphericity was performed alongside rmANOVAs for each nutrition support usage variable. If rmANOVAs were significant, then the Greenhouse–Geiser-corrected rmANOVA results were interpreted. Significant rmANOVAs were followed up with Fisher’s Least Significant Difference test to determine where each variable differed over time. Partial eta-squared values are reported as effect sizes.

## 3. Results

### 3.1. Demographics

Most parents in the sample were between the ages of 35–44 (*n* = 40), were primarily female (*n* = 66), and had a family income of less than $35,000 annually (*n* = 82). For parents who reported race (*n* = 84), over a third were African American or Black (*n* = 31). Over a third of respondents (*n* = 45) reported that someone in the family lost their job due to the COVID pandemic. Only 29.5% (*n* = 36) verified that no one in their household lost employment due to COVID (Figure 1).

### 3.2. Food Security and Federal Nutrition Program Utilization

Based on the Wilcoxon Signed-Rank test, there were no significant differences (*z* = 0.00, *p* = 1.00) in food security from pre- to post-COVID-19 despite the reduction in families reporting very low food security before and after the pandemic (Table 2).

Parents reported similar utilization rates of SNAP/WIC from T1 to T2. There were slight reductions in the number of families having to choose between food and utilities and healthy food vs. inexpensive unhealthy food from pre- to post-pandemic (Table 2).

Participation in school breakfast and lunch programs decreased by half from T0 to T2. School pantry usage remained constant and community pantry usage dropped slightly from T0 to T2 (Table 3).

Pearson correlations determined the relationship between school meal programs and food security. For reference, lower sum scores for the food security questions indicated higher food security, and higher scores for community and school resource questions indicated more frequent use. Pre-COVID, postintervention (T1) food security was not significantly correlated with any of the factors related to program use at its time point. Post-COVID, postintervention (T2) food security had a negative relationship with eating breakfast at home and a positive relationship with utilizing a community food pantry. Other relationships for post-COVID, postintervention food security included a negative correlation with eating breakfast at home, with breakfast at school, or lunch at school. Additionally, eating breakfast or lunch at school was positively correlated with utilizing food pantries at school and in the community (Table 4).

The use of two school programs and breakfast consumption at home were found to be significantly different over time (Table 5); the school pantry and community pantry use did not change significantly over time. Breakfast consumption nonsignificantly (*p* = 0.28) decreased from T0 (*M* = 4.11) to T1 (*M* = 3.91) but significantly (*p* = 0.01) increased from T1 to T2 (*M* = 4.36); however, T2 breakfast consumption was not significantly different (*p* = 0.12) from T0. Consumption of school breakfast significantly (*p* = 0.007) decreased from T0 (*M* = 2.90) to T1 (*M* = 2.38) and significantly (*p* < 0.001) to T2 (*M* = 1.47); thus, there was an overall 1.42 significant (*p* < 0.001) decrease in breakfast consumption from T0 to T2. Finally, school lunch consumption also significantly (*p* = 0.03) decreased from T0 (*M* = 3.50) to T1 (*M* = 3.11) and significantly (*p* < 0.001) to T2 (*M* = 1.80); thus, there was a significant (*p* < 0.001) overall 1.70 decrease in school lunch consumption from T0 to T2.

### 3.3. Eating Behaviors

Fruit and vegetable consumption were assessed to determine eating behaviors. There were typically at least two or more servings of fruit per person per day in 64.4% of households and three or more servings of vegetables per person per day in 23.1% of households. Parents averaged consuming fruit 1.39 times daily at T2 (19.8% met the Recommended Dietary Intake (RDI)) and reported 1.62 servings at T0 (preintervention, pre-COVID; 20.0% met RDI) and 1.44 at T1 (postintervention, pre-COVID; 19.3% met RDI)). Vegetable consumption followed a similar pattern; at T2, parents reported 1.95 servings of vegetables (11.3% met RDI), with 2.34 servings at T0 (17.2% met RDI) and 2.24 servings at T1 (12.3% met RDI). Fruit intake did not significantly differ over time (*F*(2, 148) = 1.26, *p* = 0.29), nor did total vegetable intake (*F*(1.56, 142) = 2.55, *p* = 0.095); vegetable intake violated the assumption of sphericity and a Greenhouse–Geisser correction was used (*χ^2^*(2) = 23.07, *p* < 0.001).

Parents rated their own health and eating habits positively (Figure 2). Many parents also reported that their eating habits changed for the better at T1 (38.8%, *n* = 47). Fifteen parents stated that their eating habits changed for the worse following the onset of COVID-19, yet 83.7% of parents stated they wanted to improve their family’s eating habits. In addition, 32% of parents reported that they would like to set a goal to assist their child in eating more fruits and vegetables “about every day” and 81.8% would want assistance helping their child set goals for fruits and vegetables half of the time or more.

## 4. Discussion

Evidence demonstrates that behavior-focused interventions tend to have the largest positive influence on healthy eating habits [65]. However, behavior change remains difficult when food insecurity is a reality for those who have been disproportionately impacted by uncoordinated policies and systems [40]. The COVID-19 pandemic impacted many aspects of individuals’ lives, including healthy food access and consumption [5,43,66,67]. This was evident among the BFF sample, as half of participants continued to report usage of state and federal programs during both measured time points throughout the pandemic (T1 and T2). Additionally, a considerable number of parents in this study reported a loss of employment in their family during the COVID-19 pandemic, which has been shown to drastically influence their ability to keep their families fed [68].

There was an increase in demand for food aid during the COVID-19 pandemic across the U.S., particularly among the midwestern communities involved in the BFF intervention [5,46]. Although there was no statistically significant change in food security status, food insecurity reported by families did decrease over time. This may be the result of increased usage of school and community food programs, which was significant as many families in this sample reported high or marginal food security following the onset of COVID-19. The correlations displayed in Table 2 overall support more frequent usage of school or community programs if families used at least one other program.

At T2, those with lower levels of food security were less likely to eat breakfast at home and more likely to obtain food from a community pantry. Since these families were less likely to eat breakfast at home, it is interesting that there was not a significant correlation of lower food security with higher use of school breakfast programs. Additionally, breakfast consumption at home increased significantly from T1 to T2. This could mean that families with lower food security status went without breakfast at all, indicating the need to focus on giving families better access to school and community breakfast programs. The results of this study also illustrate a need to understand the utilization of federal interventions among this and other low-income food-insecure populations, which may have been the reason that some BFF families experienced a decrease in food insecurity despite job and income loss. In March 2020, P-EBT was created at the federal level, which increased financial assistance and also increased the monetary SNAP benefits to low-income families with school-aged children during the COVID-19 pandemic [69]. Overall, SNAP participation assists in lowering food insecurity, yet some food-insecure families still may not utilize these benefits [69].

Significant decreases in the use of school breakfast and lunch programs among BFF families are likely explained by the limited interactions with the school building that families experienced post pandemic and may also be a result of additional social and financial support programs disseminated during the pandemic. Despite the great need for these programs, only about half of parents reported that their children regularly utilized school programs such as free breakfast and free lunch following the intervention and the onset of COVID-19. Barriers to program utilization during the COVID-19 pandemic need to be studied further and may have included the following: (1) safety and health concerns, (2) employment status, (3) a lack of transportation, (4) a new reliance on increased SNAP benefits via P-EBT, (5) lack of communication about school meal adaptations and pick up times, and (6) the constraints of working from home or having no childcare [70]. However, some BFF families were able to utilize bus routes and have school meals delivered to their homes for easy utilization, while other families relied on a drive-thru style meal pick-up offering. School meal program distribution during the pandemic was dependent on school district practices. Pearson Correlations further describing the relationships between school and community programs supported consistency among program usage. For example, if participants used the school breakfast program, they were likely to also use other programs such as school lunch and other supports from the school or community, such as a mobile pantry. This demonstrates that if a family had issues with access, this likely continued despite the goals of the intervention and innovation utilized within each school district.

Only one in ten adults consume enough fruits and vegetables on a daily basis, which was consistent with our results [70]. However, more than half the parents in this sample rated their health and their eating habits as good or better, despite low fruit and vegetable intake. With poor diet being a risk factor for noncommunicable and chronic diseases as well as COVID-19 hospitalization and death, inadequate fruit and vegetable intake increases the potential for developing these diseases [42]. Parents were relatively young in the sample, but 33.6% of parents reported having one or more chronic diseases at the time of the study. Neither fruit nor vegetable intake increased over any of the three time points. Interestingly, nearly all parents reported that they wanted to change their eating habits for the better despite positive ratings of eating habits. This may indicate a need for further nutrition education components within interventions to support parents in making healthy eating decisions for their families [71]. Since neither fruit nor vegetable intake increased over time, community partners—especially food banks—should work to identify the culturally relevant fruits and vegetables their school district families prefer and aim to consistently make those options more available [72,73]. In turn, the partners designing the intervention could pilot healthier food boxes that are more aligned with community supported agriculture boxes as an alternative to sourcing processed foods by utilizing USDA programs like Farm to Table or Farmers to Families [47,69].

## 5. Conclusions

There are lessons learned from this study that should be applied when implementing food access interventions in school-based settings both with and without the lens of COVID-19. First, a coordinated nutrition security approach could be adopted at the federal level and implemented at the state and local levels that capitalizes on local assets and resources to meet community needs. For example, P-EBT was found to be transformative for many BFF participants and other low-income families throughout Michigan [74]. This can be demonstrated by the evidence that food security did not change among the BFF participants despite a number of families reporting loss of income and wages. It is recommended that the federal government continue P-EBT benefits in collaboration with state partners beyond the pandemic to lift families out of poverty and provide food security for millions. Additionally, BFF families had continued access to free school meals and, for those that qualified, an increased SNAP or WIC benefit due to federal policy changes [5]. Universal school lunch and breakfast programs could easily be institutionalized in the next revision of the Child Nutrition Reauthorization. This too would assist millions of low-income families and children in meeting their nutritional needs and obtaining food security.

Second, innovations that were developed by local school districts at the height of the pandemic exhibit evidence for sustainable school health practices. Families from the two different districts had different experiences with COVID-19 and food access. In this study, some school districts in the area utilized bus routes to deliver food during the COVID-19 pandemic. It is likely that families in districts with bus routes had an advantage in utilizing school meal programs due to this innovative response. The option for bus routes should be considered when evaluating the use of school and community programs, as guidelines for implementation of federal programs differ in varying locations within each state [75]. This is especially true for school districts that operate the Seamless Summer Option or Summer Food Service Programs, recognized holidays, and planned breaks during the academic school year [56].

Third, there is more to be done to engage adults, youth, families, and the organizations they rely on for nutrition supports to discern timely, culturally relevant nutrition programs that promote the intake of fresh fruits and vegetables. The population of focus did not report any significant increases in fruit and vegetable intake even though they may have had more available healthy food options to do so. However, the lingering impacts of unemployment due to COVID-19 may have contributed to the scarcity habits of food insecurity where any calorie is better than no calorie. Additionally, across the U.S., healthy eating behaviors decreased among all populations early on in the pandemic as there were fears about spread of the virus through fresh produce and the availability of healthy foods in grocery stores was greatly reduced [76,77]. While there is still more to learn, especially among food-insecure populations, it is evident through the data of this study that further research is needed to deconstruct the PSEs that elicit a scarcity response among food-insecure families as well as more targeted efforts of nutrition education that are culturally relevant and effective.

There were a few limitations of the reported study. First, the participants reported the number of servings of fruits and vegetables instead of number of cups. This could overestimate the number of fruits or vegetables actually being consumed [60], as individuals may not know what is considered a serving size. Parents also reported availability of fruits and vegetables in their households using serving sizes as the measure. Second, the data analyzed were self-reported retrospective survey data; therefore, there may have been some biases in the responses recorded and analyzed. Finally, COVID-19 placed limitations on the methods that had been initially planned for the evaluation. For example, the recruitment and sampling methods changed, which resulted in fewer student families in the study activities and an inability to study the families over time. Additionally, data were collected over several months among the parent participants due to contact limitations and health fears. The relationships and changes over time were likely affected by the within-time-period measurement lags. With shorter measurement periods and/or larger samples, to enable a more complex analysis model that included individuals’ time of measurement as a lag variable [78,79], clearer time-specific effects of future interventions may be captured and will assist in generalizing the results of the BFF evaluation for future research reports.

This manuscript contributes to the literature via reporting on the results of an evaluation of an intervention interrupted by the COVID-19 pandemic while providing lessons learned for public health practice. Future research should examine the impact of food security interventions on the youth–parent dyads present in this study. Looking at dyads would enable a deeper sense of how to construct more equitable nutrition support policies, and the systems and environments they are disseminated through to alleviate the generational impacts of food insecurity on families. Understanding the financial burden of nutrition support interventions that resulted from COVID-19 could also help inform a coordinated and strategic approach to federally funded programs. The design of and dissemination of COVID-19-related nutrition supports has had unintended financial consequences on school districts whose school-aged youth are dependent on a typical school year. Since our population of focus showed a downward trend in utilization of school food programs, future research is needed. Economic simulations that can retrospectively discern alternative dissemination plans for programs, such as the simultaneous dispensation of P-EBT and free school meals, might enable researchers to inform sustainable approaches to normalize both programs across school districts nationally. It is known that behavioral-based interventions that focus on environmental factors may be most beneficial to implement in the school setting. Following the onset of COVID-19, many families experienced food insecurity but have received support from local, state, and federal interventions. Additionally, many uncertainties remain as a result of the COVID-19 pandemic and its larger impact on low-income, marginalized families.

## Figures and Tables

**Figure 1 ijerph-18-11006-f001:**
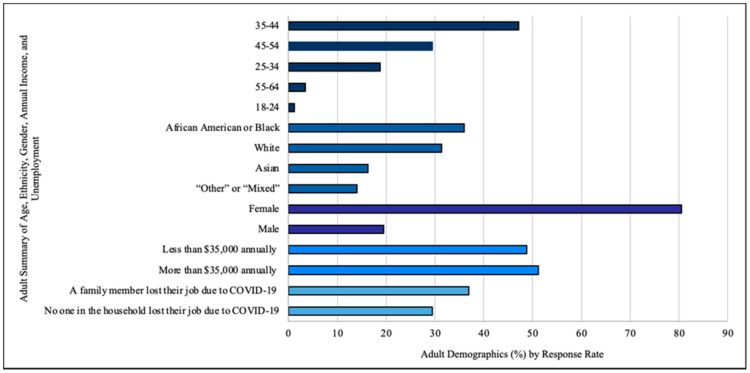
Parent Demographics.

**Figure 2 ijerph-18-11006-f002:**
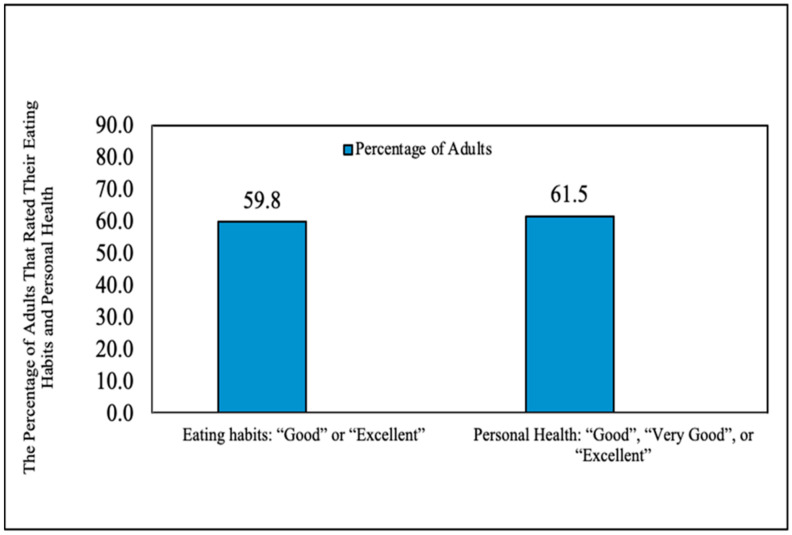
Parents ratings of eating habits and personal health.

**Table 1 ijerph-18-11006-t001:** Socioeconomic status indicators among adults by ethnicity and race in the United States. Reproduced from references [13,14].

Race/Ethnicity	Unemployment Rate (2nd Half 2020)	Adult Poverty Rate (2019)	Projected Poverty Rates (2020)	Projected Poverty Rates (2020) without COVID-19 Related Financial Responses	Child Poverty Rate (2020)
Hispanic/Latino	16.7%	18.8%	13.7%	18.2%	29.2%
Non-Hispanic Black	16.1%	15.7%	15.2%	20.5%	27.3%
Asian/Pacific Islander	14.3%	7.3%	8.1%	10.6%	9.3%
Non-Hispanic/White	12.0%	9.1%	6.6%	9.0%	9.1%

**Table 2 ijerph-18-11006-t002:** Rates of Food Security and Federal Nutrition Program Participation.

	Pre-COVID-19 Onset	Post-COVID-19 Onset
Very Low Food Security	12.3%	8.2%
Use SNAP/WIC Benefits	51.1%	50.0%
SNAP/WIC Use 1+ years	61.4%	52.8%
Chose between food and utilities	10.6%	8.2%
Chose between healthy food and inexpensive, unhealthy food	13.1%	11.5%

**Table 3 ijerph-18-11006-t003:** Frequency of Nutrition Support Use by Time Point.

	T0	T1	T2
Breakfast at school 3–5 days/week	61.6%	52.4%	27.6%
Lunch at school 3–5 days/week	76.6%	68.4%	38.2%
School pantry 2–4 times/month	31.6%	29.8%	29.3%
Community pantry 2–4 times/month	24.5%	18.6%	18.6%

**Table 4 ijerph-18-11006-t004:** Pearson correlations with 95% CI for program use and food security variables at each time point.

	T0	T1	T2
Home Breakfast and School Breakfast	−0.18 −0.37, 0.03	−0.08 −0.30, 0.14	−0.24 * −0.41, −0.04
Home Breakfast and School Lunch	−0.23 * −0.41, −0.02	0.084 −0.14, 0.30	−0.22 * −0.40, −0.02
Home Breakfast and After-School Program	0.03 −0.18, 0.24	−0.03 −0.24, 0.19	N/A
Home Breakfast and School Pantry	0.14 −0.07, 0.34	−0.16 −0.36, 0.06	−0.03 −0.22, 0.18
Home Breakfast and Community Pantry Use	0.03 −0.18, 0.24	−0.16 −0.36, 0.06	−0.055 −0.25, 0.15
School Breakfast and School Lunch	0.53 ** 0.36, 0.66	0.66 ** 0.51, 0.77	0.80 ** 0.71, 0.86
School Breakfast and After-School Program	0.473 ** 0.29, 0.62	0.68 ** 0.54, 0.78	N/A
School Breakfast and School Pantry Use	0.29 ** 0.09, 0.47	0.32 ** 0.11, 0.50	0.41 ** 0.23, 0.56
School Breakfast and Community Pantry Use	0.29 ** 0.08, 0.47	0.28 * 0.07, 0.47	0.46 ** 0.28, 0.60
School Lunch and After-School Program	0.35 ** 0.15, 0.52	0.44 ** 0.24, 0.60	N/A
School Lunch and School Pantry Use	0.13 −0.08, 0.33	0.13 −0.10, 0.33	0.46 ** 0.29, 0.61
School Lunch and Community Pantry Use	0.08 −0.14, 0.28	0.14 −0.08,.35	0.33 ** 0.14, 0.50
After-School Program and School Pantry Use	0.47 ** 0.29, 0.62	0.39 ** 0.19, 0.56	N/A
After-School Program and Community Pantry Use	0.52 ** 0.34, 0.66	0.55 ** 0.37, 0.68	N/A
School Pantry Use and Community Pantry Use	0.70 ** 0.57, 0.79	0.65 ** 0.51, 0.76	0.51 ** 0.34, 0.64

Note: Lower scores for food security questions indicate higher food security; higher scores for community and school resource questions indicate more frequent use. * Denotes significance at *p* = 0.05. ** Denotes significance at *p* = 0.01.

**Table 5 ijerph-18-11006-t005:** Repeated-measures ANOVA omnibus results with Mauchly’s test of sphericity.

	Mauchly’s Test (*χ^2^*(2))	ANOVA Result *F*(df1, df2)	Effect Size (Partial *η^2^*)
Breakfast consumption at home	2.66	(2.83) 3.46 *	0.04
Breakfast consumption at school	9.82 **	(1.78, 74) 19.64 **	0.21
Lunch consumption at school	29.52 **	(1.51, 74) 23.30 **	0.24
School pantry use	3.66	(2.80) 0.27	0.003
Community pantry use	20.38 **	(1.53, 78) 1.77	0.04

* Denotes significant result at 0.05. ** Denotes significant result at 0.01.

## Data Availability

Due to the nature of this research, participants of this study did not agree for their data to be shared publicly, so supporting data is not available.

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
