# Peer review of "Nutrition Supports Deconstructed and Disrupted: An Evaluation of a Multilevel School-Based Intervention during the Time of COVID"

_ijerph, 2021, doi:10.3390/ijerph182111006_

Round 1

Reviewer 1 Report

Important topic

Author Response

Dear Reviewer 1:

Thank you for your time and attention in reviewing this manuscript. We believe we have addressed all of the comments on our manuscript and have provided a specific response to each comment below.

Comment 1: Recommend including a formal definition of the term.

Response: Included a definition of food insecurity on lines 42-43.

Comment 2: Beginning on line 51, revise for clarity the link between lines 51-54.

Response: Revised and clarified the link between food security and PSEs on lines 48-53

Comment 3: Line 66: add information to subsequent

Response: Added information to lines 68-70

Comment 4: Be sure that the headings match the order of content for ease of reading. E.g. paragraph on line 81 should be presented first under this heading based on the content sequence in the heading.

Response: Adjusted the title of this heading on line 75

Comment 5: Spelling error line 87, multiplies vs. multiples

Response: Fixed this. It is now on line 91.

Regards,

Manuscript First Author

Reviewer 2 Report

Comments to the authors

Line 26: please report in a clearer way data on income.

In general, subtitles of the introduction are not accepted. I suggest to Delete them.

Lines 62-64: data on unemployment do not match figures in the table: it is not possible that the minimum of the variable measured by race is higher than the maximum of the same variable measured by gender.

Line 105: misspelling of susce…

Lines 104-112: reference 39: there are many, more established papers on the subject of covid, health and food security. Please include at least some of them (e.g. Klassen and Murphy, 2020; Ekumah et al., 2020).

Lines 148-153: please cite estimates from official sources. Moreover, 49: seems not a valid references for such data

Line 160: misspelling: "solutions"?

Lines 205:216: from this description it is not clear if respondents in both schools have been interviewed in the 3 time frames

Lines 243-245: since the authors explain better how they have categorised the variable later in the paper, these lines are not useful.

Lines 247-248: which “cleaning” procedure has been applied to data? To which variables? Why? How?

Lines 270-283: from this paragraph, it seems that missing values are at least a third of the answers. Perhaps it would have been better to present a table with demographic figures at the 3 times (including both answers and the amount of missing values).

Section 3.2: please present at least a table including average (mean) values of the variables of interest at the 3 time points.

Table 2: not very clear: mean difference of the frequency? Please explain better.

Table 3: it is not very clear to me between which variable are calculated the correlation coefficients?

In the table, the authors can use either stars or p-values, since they provide the same information.

Discussion: Not all the information presented are related to the results of the analysis. For instance, in lines 382-385 the observation provided are related to the cited reference but no evidence of this is provided in the paper.

Conclusions: some very general opinions are provided. I suggest to focus much more on the more specific results of the study.

Line 484: limitations of the study are at least "some". Even if a test is provided, nor the sample size nor its composition allow for such general conclusion.

Author Response

Thank you for your time and attention in reviewing this manuscript. We believe we have addressed all of the comments provided and have included a specific response to each comment below.

Comment 1: Line 26: please report in a clearer way data on income.

Response: Adjusted income reporting on line 26.

Comment 2: In general, subtitles of the introduction are not accepted. I suggest to Delete them.

Response: The authors feel as though the subtitles assist with flow of the introduction and the many components of the paper we need to introduce. We have kept the subtitles but shortened the introduction.

Comment 3: Lines 62-64: data on unemployment do not match figures in the table: it is not possible that the minimum of the variable measured by race is higher than the maximum of the same variable measured by gender.

Response: We have clarified in the title of the table that all adults are being reported (not only females or males who lost employment). This is on lines 72-73. 

Comment 4: Line 105: misspelling of susce…

Response: Corrected misspelling on line 110.

Comment 5: Lines 104-112: reference 39: there are many, more established papers on the subject of covid, health and food security. Please include at least some of them (e.g. Klassen and Murphy, 2020; Ekumah et al., 2020).

Response: Added these references. Line 109.

Comment 6: Lines 148-153: please cite estimates from official sources. Moreover, 49: seems not a valid references for such data

Response: Added additional references from the U.S. Census Bureau and the Michigan Department of Health and Human Services on lines 151-162. Removed the previous reference 49 as requested.

Comment 7: Line 160: misspelling: "solutions"?

Response: Removed this language from the manuscript.

Comment 8: Lines 205:216: from this description it is not clear if respondents in both schools have been interviewed in the 3 time frames.

Response: Clarified the evaluation data collection periods and sampling lines 202-220.

Comment 9: Lines 243-245: since the authors explain better how they have categorised the variable later in the paper, these lines are not useful.

Response: Clarified the instrument variables and scoring, lines 222-259.

Comment 10: Lines 247-248: which “cleaning” procedure has been applied to data? To which variables? Why? How?

Response: Clarified how the data was cleaned and prepared for analysis, lines 261-265.

Comment 11: Lines 270-283: from this paragraph, it seems that missing values are at least a third of the answers. Perhaps it would have been better to present a table with demographic figures at the 3 times (including both answers and the amount of missing values).

Response: Missing values were removed from the text and a Figure was added, lines 280-288.

Comment 12: Section 3.2: please present at least a table including average (mean) values of the variables of interest at the 3 time points.

Response: Two tables were added for clarity. Lines 294-295 and 303-305. 

Comment 13: Table 2: not very clear: mean difference of the frequency? Please explain better.

Response: Now Table 5 in the manuscript this was revised and clarified. Lines 340-342.

Comment 14: Table 3: it is not very clear to me between which variable are calculated the correlation coefficients? In the table, the authors can use either stars or p-values, since they provide the same information.

Response: Now Table 4 in the manuscript this was revised and clarified. Lines 322-323.

Comment 15: Discussion: Not all the information presented are related to the results of the analysis. For instance, in lines 382-385 the observation provided are related to the cited reference but no evidence of this is provided in the paper.

Response: This information was removed from the manuscript. Language was softened in terms of policy recommendations and connected more thoroughly to the results of the study. Lines 369-439. 

Comment 16: Conclusions: some very general opinions are provided. I suggest to focus much more on the more specific results of the study.

Response: Conclusion was revised and recommendations were tied to the results of the study. Lines 441-518.

Comment 17: Line 484: limitations of the study are at least "some". Even if a test is provided, nor the sample size nor its composition allow for such general conclusion.

Response: Limitations of the study were clarified. Lines 481-497.

Reviewer 3 Report

This is a very interesting and timely paper. I think that this paper is incredibly relevant to the special issue and the interests of IJERPH. I think this paper has real merit and importance to the literature, which is why I made a lot of comments, but know that these are being made to improve the impact of the paper and reflect the importance of this work. So do not be disheartened by my comments/questions. Please see the comments below organized by section of the manuscript -  

  • Overall - 
    • Make sure that the purpose of the paper (identified on lines 55-58, and 198-202) align with what is presented in the paper. I think that a more accurate representation of the paper is that it introduces and explains the BFF program, BFF evaluation, and presents baseline survey data with an emphasis on food insecurity, food assistance and school meal utilization, and nutrition outcomes during COVID (timing is a bit confusing with the survey, and various timepoints). 
  • Introduction/Background - 
    • The flow is a bit choppy, please consider restructuring the introduction. I believe that the purpose of the paper is potentially introduced too early on in the intro and that is contributing to the lack of flow. 
    • Also, the introduction, while very well-researched, feels a bit too long. Consider streamlining some sections for a more succinct introduction. 
    • Table 1: Please add what the population is, are these national rates, or more specific to your region of interest? Based on your citations (7-10, 14) it is not clear. Also, since you are using multiple sources, are these data comparable? If not, consider integrating the data presented in this table into a paragraph.
    • I'm a bit confused about the initiation of the BFF program related to COVID -- it is an 8-10 year initiative, but did it start prior to the pandemic? adding years in the Intro could help orient the methods better. 
    • Minor comment - Please add sources for the sentence on lines 53-54
  • Methods - 
    • See comment above about the initiation of the BFF program. This will help better orient if the term "baseline" is appropriate. 
    • More information is needed about the selection criteria  and recruitment strategies used for SD1 and SD2 - are these districts comparable socioeconomically? How do you account for the fact that SD1 was recruited prior to (or at the start) of when the COVID-19 pandemic first hit Wayne County? Having a consort diagram could be helpful with this. 
    • Add clarifying language around when the survey was administered and the various timepoints. It is a bit confusing if the timepoints are referring to timepoints asked about in the one baseline survey, or if the survey was administered at 3 timepoints. 
    • Is this modified version of the food security measure validated? 
    • Potentially consider adding a paragraph at the beginning of the methods section of the overall evaluation plan. I understand that you are only presenting baseline survey findings (which is great), but having context as to how this piece fits in to the larger evaluation may be helpful  for the reader (and you can cite this for future publications of other evaluation components). This could also help orient the reader as to how T1-T3 related to COVID and the intervention implementation. 
  • Results - 
    • All the results presented were just from the parent survey, correct? If not, clearly delineate whether the youth or parents answered each section. 
    • Consider making a table for the findings presented in section 3.1 and 3.2 all of the parentheses are a bit jarring to the reader.
    • Could the authors expand on why food insecurity was not asked at T1? Same question for SNAP and WIC.
    • Please refer to the Tables 2 and 3 in the paragraphs that you are presenting the findings of. 
    • Table 3 is a bit confusing for what is the correlation and what is the p-value, consider reformatting. Also are these correlations for food security and program use, or just program use? It's a bit confusing. 
    • Eating behavior results narrative is a bit confusing - potentially present findings pre-intervention, during intervention, and then post-intervention. Using phrasing like "at the time of the survey" and then pre/post intervention is just a bit confusing. 
  • Discussion - 
    • The paragraph from lines 386-393 is a bit confusing and appears to contradict itself.
    • Some of the statements in the discussion are a bit of an extrapolation from the findings. Potentially say "future research should explore..." rather than hypothesizing reasons that are not directly related to the sample and baseline survey (like in lines 411-415). 
  • Conclusion - 
    • Nice summary! However, potentially streamline this section a bit. 
    • For the limitations, please include a discussion of possible threats to validity/biases - including a discussion of potential external validity. 

I know those are a lot of comments, but I think this paper, once restructured and edited for clarity, will be a valuable contribution to the literature. 

Author Response

Thank you for your time and attention in reviewing this manuscript. We believe we have addressed all of the comments provided and have included a specific response to each comment below.

Comment 1: Make sure that the purpose of the paper (identified on lines 55-58, and 198-202) align with what is presented in the paper. I think that a more accurate representation of the paper is that it introduces and explains the BFF program, BFF evaluation, and presents baseline survey data with an emphasis on food insecurity, food assistance and school meal utilization, and nutrition outcomes during COVID (timing is a bit confusing with the survey, and various timepoints). 

Response: Purpose of the paper was revised and included on lines 194-199.

Comment 2: The flow is a bit choppy, please consider restructuring the introduction. I believe that the purpose of the paper is potentially introduced too early on in the intro and that is contributing to the lack of flow. Also, the introduction, while very well-researched, feels a bit too long. Consider streamlining some sections for a more succinct introduction. 

Response: Introduction was revised and shortened, purpose of the paper was only included on lines 194-199.

Comment 3: Table 1: Please add what the population is, are these national rates, or more specific to your region of interest? Based on your citations (7-10, 14) it is not clear. Also, since you are using multiple sources, are these data comparable? If not, consider integrating the data presented in this table into a paragraph.

Response: Clarified title of table and references. Line 72.

Comment 4: I'm a bit confused about the initiation of the BFF program related to COVID -- it is an 8-10 year initiative, but did it start prior to the pandemic? adding years in the Intro could help orient the methods better. 

Response: Years added to introduction of BFF lines 169-170.

Comment 5: Please add sources for the sentence on lines 53-54.

Response: References added. Line 59.

Comment 6: See comment above about the initiation of the BFF program. This will help better orient if the term "baseline" is appropriate. 

Comment 7: More information is needed about the selection criteria  and recruitment strategies used for SD1 and SD2 - are these districts comparable socioeconomically? How do you account for the fact that SD1 was recruited prior to (or at the start) of when the COVID-19 pandemic first hit Wayne County? Having a consort diagram could be helpful with this. 

Comment 8: Add clarifying language around when the survey was administered and the various timepoints. It is a bit confusing if the timepoints are referring to timepoints asked about in the one baseline survey, or if the survey was administered at 3 timepoints. 

Comment 9: Potentially consider adding a paragraph at the beginning of the methods section of the overall evaluation plan. I understand that you are only presenting baseline survey findings (which is great), but having context as to how this piece fits in to the larger evaluation may be helpful  for the reader (and you can cite this for future publications of other evaluation components). This could also help orient the reader as to how T1-T3 related to COVID and the intervention implementation. 

Response: The methods section was revised and clarified to include the overall evaluation plan and the description of the parent survey collection time points within the baseline assessment. Lines 202-220.

Comment 10: Is this modified version of the food security measure validated? 

Response: References added for validation of modification of USDA Adult Food Security measure. Line 255.

Comment 11: All the results presented were just from the parent survey, correct? If not, clearly delineate whether the youth or parents answered each section. 

Response: Sample and study participant data that is being reported in this manuscript was clarified. References to the youth survey were removed as only the parent data is being reported at this time. Lines 219-220 and 280-285.

Comment 12: Consider making a table for the findings presented in section 3.1 and 3.2 all of the parentheses are a bit jarring to the reader.

Response: Tables and figures added to results. Lines 286-205.

Comment 13: Could the authors expand on why food insecurity was not asked at T1? Same question for SNAP and WIC.

Response: The parent survey was revised in June 2020 after the onset of the COVID-19 pandemic. As this was meant to be a baseline assessment (with an additional two time points collected in Fall 2020 and Winter 2021) we only captured food insecurity and SNAP/WIC before and after COVID-19. The delays and stoppages in data collection due to the pandemic did not allow for additional measures in the BFF evaluation. This was clarified on lines 225-230 where we discuss the instrument.

Comment 14: Please refer to the Tables 2 and 3 in the paragraphs that you are presenting the findings of. 

Response: Table citations included in text. Lines 292, 298, 301, 316, 329.

Comment 15: Table 3 is a bit confusing for what is the correlation and what is the p-value, consider reformatting. Also are these correlations for food security and program use, or just program use? It's a bit confusing.

Response: Now Table 4 in the manuscript was revised and clarified. Lines 318-323. 

Comment 16: Eating behavior results narrative is a bit confusing - potentially present findings pre-intervention, during intervention, and then post-intervention. Using phrasing like "at the time of the survey" and then pre/post intervention is just a bit confusing. 

Response: Eating behavior narrative was revised and clarified. Time points for retrospective data was provided. Lines 345-355.

Comment 17: The paragraph from lines 386-393 is a bit confusing and appears to contradict itself.

Comment 18: Some of the statements in the discussion are a bit of an extrapolation from the findings. Potentially say "future research should explore..." rather than hypothesizing reasons that are not directly related to the sample and baseline survey (like in lines 411-415). 

Response: Language was softened in terms of policy recommendations and connected more thoroughly to the results of the study. Future research projects were recommended. Lines 369-439.

Comment 19: Nice summary! However, potentially streamline this section a bit. 

Response: Conclusion was revised and recommendations were tied to the results of the study. Lines 441-518.

Comment 20: For the limitations, please include a discussion of possible threats to validity/biases - including a discussion of potential external validity. 

Response: A discussion of threats to validity and external validity was included in the limitations. Lines 481-497.

Regards,

Manuscript Authors

Round 2

Reviewer 2 Report

The updated manuscript is much improved on the previous versions. I am pleased that the authors have managed to get the input of an expert statistician to help improve their paper and found my previous comments useful.

However, I still have doubts on the figures in Table 1. I have found different figures here:https://www.bls.gov/web/empsit/cpsee_e16.htm

14 is not a reference for unemployment.

Author Response

Thank you for reviewing our manuscript for a second round. Please see our specific changes in this version of the manuscript noted below and in track changes within the document.

Reviewer 2 Comment:

The updated manuscript is much improved on the previous versions. I am pleased that the authors have managed to get the input of an expert statistician to help improve their paper and found my previous comments useful.

However, I still have doubts on the figures in Table 1. I have found different figures here:https://www.bls.gov/web/empsit/cpsee_e16.htm

14 is not a reference for unemployment.

Response: We have updated the unemployment data within the table to reflect the 2nd half of 2020 data on this website. We were previously citing the Household Pulse data from the U.S Census, there was an incorrect link within citation 14 that we have fixed.

Reviewer 3 Report

Excellent revisions to this manuscript! Please see below for minor comments: 

  • Correct column headings in Table 3 to T1-T3 rather than B1-B3, or orient the reader as to what B1-B3 means. 
  •  

Author Response

Thank you for reviewing our manuscript for a second round. Please see our specific changes in this version of the manuscript noted below and in track changes within the document.

Reviewer 3 Comment:
Excellent revisions to this manuscript! Please see below for minor comments:

Correct column headings in Table 3 to T1-T3 rather than B1-B3, or orient the reader as to what B1-B3 means.

Response: We have corrected Table 3 to reflect the T0-T2 timepoints referenced throughout the manuscript.